# Liquid Dino: A Multi-Task Neural Network towards Autonomous Driving

## Abstract

In the realm of advanced driver-assistance systems (ADAS) and autonomous driving, the accurate classification of driver emotions, behaviors and contextual environments is critical for enhancing vehicle safety and user experience. This study investigates the performance of various neural network architectures across four distinct classification tasks: Emotion Recognition, Driver Behavior Recognition, Scene-Centric Context Recognition and Vehicle-Based Context Recognition, all of which incorporate visual information captured through cameras. By utilizing camera-based data, we aim to evaluate how different neural architectures handle visual inputs in these diverse contexts, thereby exploring the robustness and generalization of each model to different real-world scenarios. We compare the performance of several state-of-the-art models and introduce a novel contribution that significantly improve classification accuracies in all areas. Our results demonstrate that the proposed Liquid Dino architecture achieves an overall average accuracy of 83.79%, outperforming other models in recognizing driver emotions, behaviors and contextual scenarios. These enhancements underscore the potential of our proposed methods in contributing to the development of more reliable and responsive ADAS.

## 1 Introduction

With the rapid advancement of intelligent transportation systems, the necessity for highly sophisticated driver monitoring systems has become increasingly paramount. Accurate detection and classification of driver emotions (Naqvi et al. (2020); Zepf et al. (2020)), behaviors (Chatziloizos et al. (2024); Shahverdy et al. (2020)) and surrounding and vehicle contexts (Worrall et al. (2012)) are essential for the development of robust Advanced Driver-Assistance Systems (ADAS) and fully autonomous vehicles. These systems not only play a critical role in enhancing road safety but also contribute significantly to improving the overall driving experience by preemptively identifying and mitigating potential hazards.

However, the intricate nature of human emotions and behaviors, combined with the ever-changing dynamics of driving environments, presents substantial challenges for conventional machine learning models. Traditional approaches, such as ResNet (He et al., 2015), VGG (Simonyan & Zisserman, 2015) and MobileNet (Howard et al., 2017a), often struggle to capture these nuanced variations with the precision required for real-world applications. This limitation underscores the need for more advanced neural network architectures capable of addressing the complex demands of modern driver monitoring systems.

In this study, we aim to overcome these challenges by introducing Liquid Dino and evaluating the performance of several state-of-the-art neural networks across four critical classification tasks: Emotion Recognition, Driver Behavior Recognition, Scene-Centric Context Recognition and Vehicle-Based Context Recognition. We used the AIDE dataset (Yang et al., 2023), from which we used the frames from each video sequence in order to classify simultaneously all the previous tasks. Also, we benchmark these tasks using both established models and our novel contributions, which include a combination of CNN with Closed-form Continuous-time Neural Networks (CFC) (Hasani et al., 2022), DINOv2 (Oquab et al., 2023) and a specialized CNN configuration applied in driver's behaviour identification before (Chatziloizos et al., 2024). These models incorporate cutting-edge

techniques designed to enhance feature extraction and classification accuracy, particularly in the context of multi-task learning.

The experimental results underscore the effectiveness of our proposed models, particularly the Liquid Dino architecture. Liquid Dino achieved the highest average accuracy of 83.79% surpassing by at least 5% previous literature and outperforming traditional models in all four tasks. Specifically, it excelled in Traffic Context Recognition with an accuracy of 95.03% and Vehicle Condition Recognition at 84.76%, setting a new benchmark in the field. This significant performance gain highlights the ability of our models to leverage advanced neural network methodologies, offering improved accuracy and efficiency in comparison to conventional approaches.

In the subsequent sections, first we present the related work and then we provide a detailed overview of the dataset, methodologies and neural network architectures employed in this study. We also present a comprehensive analysis of the results, emphasizing the substantial improvements delivered by our contributions and discussing their implications for future research in the domain of intelligent driver monitoring systems.

## 2   RELATED WORK

In earlier efforts to develop driver monitoring systems, traditional methods relied heavily on hand-crafted features and rule-based algorithms to detect driver behaviors. These systems often utilized basic computer vision techniques such as edge detection, skin tone analysis, or head pose estimation to identify potential distractions (O'Mahony et al., 2020; Dalal & Triggs, 2005; Murphy-Chutorian et al., 2007; Veeraraghavan et al., 2007). For example, detecting eye movements and blink rates was a common approach for determining driver fatigue (Jiao et al., 2014). However, these rule-based methods struggled to generalize across diverse environments, lighting conditions and driver behaviors. Additionally, they lacked the adaptability and accuracy needed for complex driving scenarios, leading to the emergence of deep learning approaches, which have since become the dominant framework for addressing these challenges.

In the context of driver monitoring systems, network architectures are typically designed with the practical constraints of deployment in on-road vehicles. Leveraging advances in deep learning, many approaches have adopted classical models that are well-established in the field. These include widely accepted architectures such as the AlexNet (Krizhevsky et al., 2012), GoogleNet (Szegedy et al., 2014), PP-Res18 (Zhou et al., 2017), VGG and ResNet families. In parallel, the need for resource-efficient solutions has led to the adoption of lightweight models like MobileNet V1/V2 (Howard et al., 2017b; Sandler et al., 2018) and ShuffleNet V1/V2 (Zhang et al., 2018; Ma et al., 2018), which are particularly advantageous for real-time applications in vehicles. For tasks involving video-based data, 3D-CNN models such as C3D (Tran et al., 2015), I3D (Carreira & Zisserman, 2017), SlowFast (Feichtenhofer et al., 2019), TimeSFormer (Bertasius et al., 2021) and 3D-ResNet (Hara et al., 2018) have been implemented to capture spatio-temporal features effectively.Also in previous methodologies, the ST-GCN (Yan et al., 2018) was employed to deal with the skeleton sequences via multi-level spatio-temporal graphs. Additionally, specialized network structures have been developed to address the unique patterns in driving-related data. In our work, we extensively utilize a combination of classical, lightweight and state-of-the-art (SOTA) baseline models to conduct a comprehensive set of experiments across various learning paradigms. This diverse array of models and input streams provides critical insights into selecting the most appropriate network structures for driving-aware systems.

To accommodate the complex requirements of multi-stream and multi-modal inputs in driving perception tasks, various fusion strategies have been developed. These strategies are generally categorized into data-level, feature-level and decision-level fusion. For instance, data-level fusion methods, such as those proposed by Ortega et al. (2020), merge infrared and depth frames through pixel-wise correlation, resulting in enhanced perception performance compared to unimodal approaches. Feature-level fusion commonly involves techniques like feature summation or concatenation to integrate diverse data streams. Furthermore, decision-level fusion approaches, such as those by Kopuklu et al. (2021), involve training separate models for each driver view and then combining the outputs based on similarity scores. In our research, we introduce the fusion of interior and exterior cameras in a form of mosaic, thereby improving the overall performance of the driving perception system.

## 3 DATASET

### 3.1 DATASET DESCRIPTION

The AssIstive Driving pErception (AIDE) dataset (Yang et al., 2023) is a novel and comprehensive resource designed to enhance the capabilities of vision-driven driver monitoring systems (DMS). The dataset was meticulously collected under real-world driving conditions, ensuring the authenticity and diversity of the data. While most datasets, either offer only the interior view of the cabin or just the outer envirnment, AIDE stands out due to its multi-view, multi-modal and multi-task design, offering a rich dataset that captures both the internal state of the driver and the external driving environment.

The dataset includes four distinct camera views to provide a wide range of visual information. Three external cameras are mounted on the vehicle to capture the left view, right view and front view, thus covering the traffic context. A fourth camera is installed inside the vehicle to monitor the driver's state. Each camera records video at a resolution of 1920x1080 pixels, with a frame rate of 15 frames per second (fps) and a dynamic range of 120 dB. These cameras are synchronized to ensure temporal alignment across all views, enabling a cohesive analysis of the driving scenario and driver behavior.

The dataset is extensively annotated, offering detailed labels for the driver's face, body posture and gestures, as well as the external traffic environment. Annotations span various categories which are the four tasks that we are going to classify, including:

- **Driver Emotions (DER)** : Labels include states such as *anxiety*, *peace* and *weariness*.

- **Driver Behaviors (DBR)**: Actions like *smoking*, *talking* and *dozing off* are annotated.

- **Scene-Centric or Traffic Context (TCR)**: Situations such as *traffic jams* or *smooth traffic* are documented.

- **Vehicle Conditions (VCR)**: The state of the vehicle, such as *parking* and *lane position*, is recorded.

This multi-task framework allows for a holistic analysis of the driving experience, integrating data on the driver's behavior and emotional state with external traffic conditions.

The data collection was conducted during naturalistic driving sessions, where participants were unaware of the specific data collection objectives to capture authentic driving behavior. Data was collected from multiple drivers with varied driving styles across different times and days, ensuring the dataset encompasses a wide range of driving scenarios and conditions. Overall, the AIDE dataset is a versatile and detailed dataset that can serve as a foundation for the next generation of driver monitoring systems.

### 3.2 PRE-PROCESSING

In our preprocessing pipeline as shown in Figure 1, we began by handling the individual images extracted from the videos of the AIDE dataset, carefully aligning and processing them for further analysis. Specifically, we merged four of these images into a single composite image arranged in a 2x2 grid format. This merging process not only streamlined the data but also allowed us to maintain the spatial relationships and contextual integrity of the original images. After forming these composite images, we resized them to smaller dimensions to facilitate faster processing and reduce computational load, while ensuring that the essential visual information was preserved. This resizing step was critical in balancing the trade-off between image clarity and the efficiency of subsequent model training and analysis. The preprocessing steps we implemented significantly optimized the dataset, making it more suitable for the demands of machine learning tasks, particularly those involving complex vision-based driver monitoring systems.

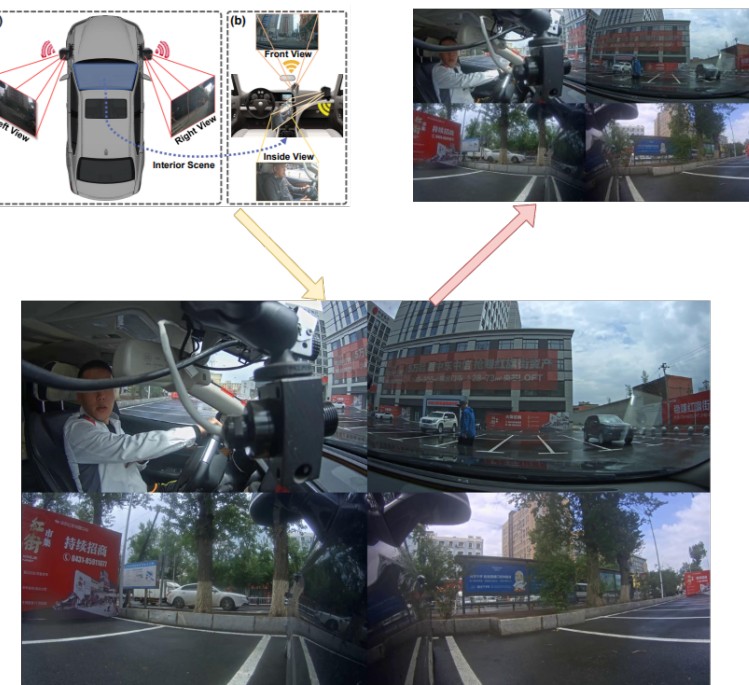

Figure 1: Data preperocessing from raw data to 2x2 re-scaled grid

## 4 METHODOLOGY

In this section, we will describe each of the methodologies used in order to develop our final model. It is important to note that the first three subsections present approaches derived from existing research, while the final subsection introduces our original Liquid Dino methodology.

Furthermore, the total loss for the classification tasks DER, DBR, TCR and VCR is calculated by summing the individual cross-entropy losses:

$$L_{\text{total}} = L_{\text{DER}} + L_{\text{DBR}} + L_{\text{TCR}} + L_{\text{VCR}}$$

Where the cross-entropy loss $L_{\text{task}}$ for each task is given by:

$$L_{\text{task}} = -\sum_{i=1}^{N} y_i \log(\hat{y}_i)$$

where $N$ is the number of classes, $y_i$ is the true label for the $i$-th class (one-hot encoded) and $\hat{y}_i$ is the predicted probability for the $i$-th class.

Thus, the total loss is:

$$L_{\text{total}} = -\left( \sum_{i=1}^{N} y_i^{\text{DER}} \log(\hat{y}_i^{\text{DER}}) + \sum_{i=1}^{N} y_i^{\text{DBR}} \log(\hat{y}_i^{\text{DBR}}) + \sum_{i=1}^{N} y_i^{\text{TCR}} \log(\hat{y}_i^{\text{TCR}}) + \sum_{i=1}^{N} y_i^{\text{VCR}} \log(\hat{y}_i^{\text{VCR}}) \right)$$

### 4.1 CNN

The Convolutional Neural Network (CNN) architecture presented in (Chatziloizos et al., 2024) was initially designed for the classification of images into 10 distinct categories, identifying the behaviour of the driver. The network is constructed using a series of MixNet-inspired blocks (Tan

& Le, 2019), each comprising convolutional layers, depthwise separable convolutions and residual connections. The initial block applies a convolutional layer with 60 channels using a 5x5 kernel, followed by batch normalization and a depthwise separable convolution with a 3x3 kernel, maintaining the same padding. Additionally, this block incorporates a Squeeze-and-Excitation (SE) (Hu et al., 2019) mechanism to bolster the network's capacity for capturing detailed and intricate patterns. This structure is iteratively refined in the subsequent blocks, where the depthwise convolutions are progressively expanded to use 5x5 and 7x7 kernels. To optimize performance, the Swish (Ramachandran et al., 2017) activation function is employed, having shown superior results compared to ReLU (Nair & Hinton, 2010) and GeLU (Hendrycks & Gimpel, 2023) in preliminary tests. The network also includes dropout layers to mitigate the risk of overfitting during training. Following the sequence of MixNet-style blocks, a final convolutional block is introduced, featuring convolutional layers with Swish activation, batch normalization and max-pooling, which collectively contribute to establishing the network's spatial hierarchy. The output layer, equipped with a softmax activation function, assigns probabilities to classify the input images into one of the specified classes. The Adam optimizer (Kingma & Ba, 2017) is utilized to fine-tune the model's parameters, aiming to maximize classification accuracy while keeping the model's complexity and parameter count to a minimum.

## 4.2 DINOV2

DINOv2 (Distillation with No Labels version 2) (Oquab et al., 2023) is an advanced self-supervised learning method designed for computer vision tasks, particularly in scenarios where labeled data is scarce or unavailable. Building on the success of its predecessor, DINOv2 leverages the power of self-distillation and Vision Transformers (ViTs) (Dosovitskiy et al., 2014) to learn rich visual representations from large-scale, unlabeled image datasets. The core innovation of DINOv2 lies in its ability to distill knowledge from a teacher model to a student model without requiring manual annotations, allowing the model to learn meaningful patterns and features directly from the data. This approach not only enhances the robustness and generalization capabilities of the model but also reduces the dependency on labeled datasets, making it highly versatile for a wide range of computer vision applications, including image classification, object detection and segmentation. DINOv2 has been shown to produce state-of-the-art results, demonstrating the effectiveness of self-supervised learning in unlocking the potential of large-scale image data.

## 4.3 CLOSED-FORM CONTINUOUS-TIME LIQUID NEURAL NETWORKS (CFC)

The hidden state of a Liquid Time-Constant (LTC) network (Hasani et al., 2022; 2020) is given by the following initial value problem (IVP), which models the system's dynamics as:

$$\frac{dx}{dt} = -\left(w_\tau + f(x, I, \theta)\right) x(t) + A f(x, I, \theta) \tag{1}$$

where $x(t)$ represents the hidden states, $I(t)$ is the input to the system, $w_\tau$ is a time-constant parameter vector, $A$ is a bias vector and $f$ is a neural network parameterized by $\theta$.

In the context of an LTC system determined by this IVP and constructed by a single cell receiving a one-dimensional time-series input $I(t)$ with no self-connections, the system's behavior can be approximated by the following closed-form solution:

$$x(t) = (x_0 - A) e^{-[w_\tau + f(I(t), \theta)]t} f(-I(t), \theta) + A \tag{2}$$

Recent advancements in continuous-time neural networks have led to a significant breakthrough with the development of such closed-form solutions, particularly within the framework of LTC networks. Traditionally, the expressive power of continuous-time models was limited by the reliance on numerical differential equation solvers. This dependency not only constrained scalability but also slowed progress in understanding complex natural phenomena, such as neural dynamics.

The introduction of tightly bounded approximations of previously unsolved integrals within the liquid time-constant dynamics has enabled the derivation of closed-form solutions, thereby circumventing the need for complex numerical solvers. This innovation has dramatically accelerated both training and inference processes by several orders of magnitude. Moreover, these advancements

allow continuous-time and continuous-depth neural models to scale more efficiently than their differential equation-based counterparts. From our experimental results, when the CFC module was used after a backbone it improved each results by around 1%.

Consequently, these closed-form models, rooted in liquid networks, have demonstrated superior performance in time-series modeling. They offer a robust alternative to advanced recurrent neural networks, marking a significant leap forward in the design of spatiotemporal decision-making systems.

## 4.4 LIQUID DINO

The Liquid Dino methodology represents a novel and sophisticated integration of the most powerful elements from CNNs, DINOv2 and closed-form continuous-time neural networks (CFC). This hybrid approach, depicted in Figure 2, is designed to leverage the unique strengths of each component model, resulting in a highly efficient and versatile framework capable of tackling complex machine learning challenges.

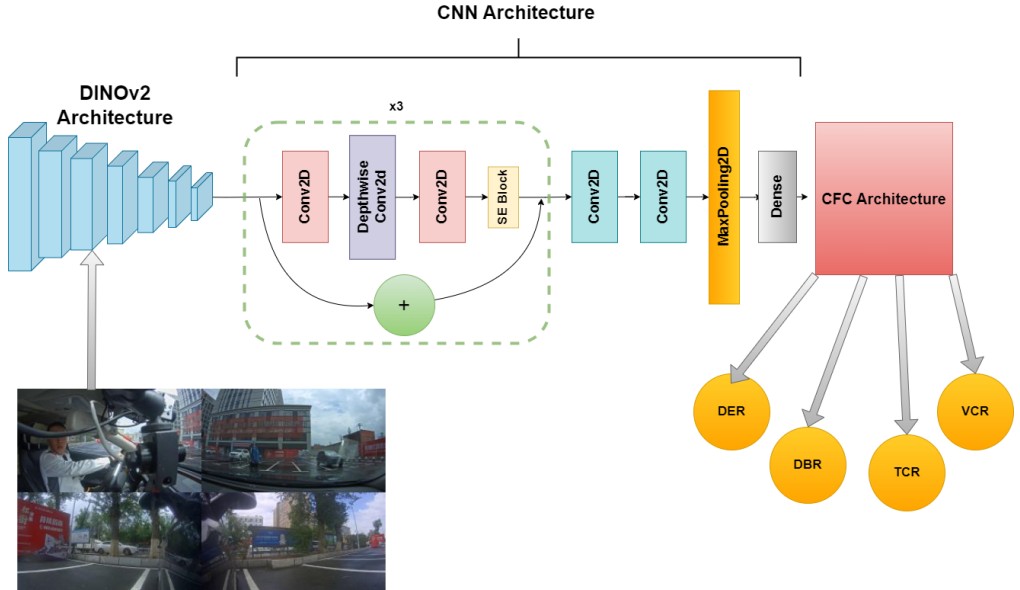

Figure 2: Liquid Dino Architecture

At its core, in the Liquid Dino methodology is DINOv2, an advanced self-supervised learning technique that excels in representation learning, especially in scenarios where labeled data is scarce or unavailable. The integration of DINOv2 into the Liquid Dino framework enhances the model's ability to discern complex patterns and structures within the data, further boosting its overall accuracy and performance.

Furthermore, Liquid Dino capitalizes on the spatial hierarchical structure and pattern recognition capabilities of Convolutional Neural Networks (CNNs). The CNN backbone is responsible for extracting rich, multi-scale features from the DINOv2 embeddings, which is particularly critical for image classification tasks where spatial relationships and local patterns are paramount. By using the CNN architecture, Liquid Dino ensures that the model captures essential low-level and high-level features that are foundational for further processing.

The final and perhaps most innovative aspect of Liquid Dino is its incorporation of closed-form continuous-time neural networks (CFC). These closed-form solutions allow Liquid Dino to process data with speed and accuracy, thus improving the overall performance of the network.

By integrating these three methodologies, CNNs for spatial feature extraction, DINOv2 for robust self-supervised representation learning and CFC, we obtain Liquid Dino, which achieves a harmo-

nious balance between accuracy, generalization and computational efficiency. This synergy not only leads to superior performance on individual tasks but also enables the model to scale effectively across various domains, making it a powerful tool in the landscape of modern machine learning.

# 5 EXPERIMENTAL RESULTS

The results of our study demonstrate significant advancements in the field of autonomous driving through the application of advanced neural network architectures for multi-task classification. We trained our models on A6000 Nvidia GPUs and tested the inference time on Qualcomm SA8255P platform which is specifically designed for automotive applications. Furthermore, we evaluated several state-of-the-art models, including the CNN + CFC, DINOv2 and CNN configurations, across four critical classification tasks: Emotion Recognition, Driver Behavior Recognition, Scene-Centric Context Recognition and Vehicle-Based Context Recognition.

## 5.1 EMOTION RECOGNITION

As shown in Table 2, the DINOv2 and the Liquid Dino model achieved the highest accuracy in recognizing driver emotions, averaging 82.93%. This model's enhanced feature extraction capabilities and robust classification framework contributed to its superior performance in distinguishing between emotions such as Anxiety, Peace, Weariness, Happiness and Anger. As shown in Figure 3, in the Emotion Confusion Matrix, the best model shows high accuracy in predicting "Peace" with significant correct predictions but struggles with distinguishing "Weariness" from "Peace" and "Anxiety."

## 5.2 DRIVER BEHAVIOR RECOGNITION

In terms of identifying driver behaviors such as Smoking, Making Phone Calls and Normal Driving, according to Table 2, the DINOv2 model demonstrated substantial improvement with an average accuracy of 72.58%. This model's architecture effectively captured subtle nuances in driver actions, enhancing its ability to classify behaviors accurately. As demonstrated in Figure 3, the Driver Behavior matrix reveals that "Normal Driving" and "Looking Around" are well-identified, whereas there is substantial confusion between "Making Phone Call" and "Normal Driving."

## 5.3 SCENE-CENTRIC CONTEXT RECOGNITION

For recognizing contextual factors like Traffic Jam, Waiting and Smooth Traffic, as shown in Table 2, the Liquid Dino configuration achieved an average accuracy of 95.06%. This configuration leveraged its multi-channel approach to effectively parse complex scenes and accurately classify driving contexts. As depicted in Figure 3, in the Scene-Centric Context matrix (TCR), "Smooth Traffic" is predominantly correctly classified, while "Traffic Jam" and "Waiting" exhibit some level of misclassification, particularly confusing "Waiting" with "Traffic Jam."

## 5.4 VEHICLE-BASED CONTEXT RECOGNITION

Across tasks involving vehicle actions such as Parking, Turning and Changing Lane, all models showed robust performance, underscoring their capability to handle diverse driving scenarios. The Liquid Dino model, in particular, excelled with its comprehensive feature representation and achieved competitive accuracy in this category. Lastly, in the Vehicle-Based Context matrix, "Forward Moving" is highly accurate, but there is noticeable confusion between "Changing Lane" and "Forward Moving."

Table 1: Pattern and Backbone models used in the previous study with corresponding IDs. The following abbreviations are used. Res: ResNet (He et al., 2015); MLP: multi-layer perception; SE: spatial embedding; TE: temporal embedding; TransE: transformer encoder (Vaswani, 2017); PP: pre-training on the Places365 dataset (Zhou et al., 2017); CG: coarse-grained.

| Pattern | ID | Backbone | | | | |
|---|---|---|---|---|---|---|
| | | Face | Body | Gesture | Posture | Scene |
| 2D | (1) | Res18 | Res34 | MLP+SE | MLP+SE | PP-Res18 |
| | (2) | Res18 | Res34 | MLP+SE | MLP+SE | Res34 |
| | (3) | Res34 | Res50 | MLP+SE | MLP+SE | Res50 |
| | (4) | VGG13 | VGG16 | MLP+SE | MLP+SE | VGG16 |
| | (5) | VGG16 | VGG19 | MLP+SE | MLP+SE | VGG19 |
| 2D + Timing | (6) | Res18+TransE | Res34+TransE | MLP+TE | MLP+TE | PP-Res18+TransE |
| | (7) | Res18+TransE | Res34+TransE | MLP+TE | MLP+TE | Res34+TransE |
| | (8) | Res34+TransE | Res50+TransE | MLP+TE | MLP+TE | Res50+TransE |
| | (9) | VGG13+TransE | VGG16+TransE | MLP+TE | MLP+TE | VGG16+TransE |
| | (10) | VGG16+TransE | VGG19+TransE | MLP+TE | MLP+TE | VGG19+TransE |
| 3D | (11) | MobileNet-V1 | MobileNet-V1 | ST-GCN | ST-GCN | MobileNet-V1 |
| | (12) | MobileNet-V2 | MobileNet-V2 | ST-GCN | ST-GCN | MobileNet-V2 |
| | (13) | ShuffleNet-V1 | ShuffleNet-V1 | ST-GCN | ST-GCN | ShuffleNet-V1 |
| | (14) | ShuffleNet-V2 | ShuffleNet-V2 | ST-GCN | ST-GCN | ShuffleNet-V2 |
| | (15) | 3D-Res18 | 3D-Res34 | ST-GCN | ST-GCN | 3D-Res18 |
| | (16) | 3D-Res34 | 3D-Res50 | ST-GCN | ST-GCN | 3D-Res34 |
| | (17) | C3D | C3D | ST-GCN | ST-GCN | C3D |
| | (18) | I3D | I3D | ST-GCN | ST-GCN | I3D |
| | (19) | SlowFast | SlowFast | ST-GCN | ST-GCN | SlowFast |
| | (20) | TimeSFormer | TimeSFormer | ST-GCN | ST-GCN | TimeSFormer |

Table 2: Accuracies of Different Models for the 4 Classification Tasks

| ID | DER | DBR | TCR | VCR | Average Acc |
|---|---|---|---|---|---|
| (1) | 69.05 | 63.87 | 88.01 | 78.16 | 74.27 |
| (2) | 71.26 | 65.35 | 83.74 | 77.12 | 74.37 |
| (3) | 69.68 | 59.77 | 80.13 | 71.26 | 70.21 |
| (4) | 70.72 | 63.65 | 82.77 | 77.94 | 73.77 |
| (5) | 69.31 | 62.34 | 83.58 | 75.13 | 72.09 |
| (6) | 70.83 | 67.32 | 90.54 | 79.97 | 77.67 |
| (7) | 72.65 | 67.08 | 86.63 | 78.46 | 76.71 |
| (8) | 70.24 | 63.54 | 82.57 | 73.69 | 72.51 |
| (9) | 71.12 | 67.15 | 85.13 | 78.58 | 75.99 |
| (10) | 69.46 | 65.48 | 85.74 | 77.91 | 74.15 |
| (11) | 72.23 | 64.20 | 88.34 | 77.83 | 75.65 |
| (12) | 68.47 | 61.74 | 86.54 | 78.66 | 73.35 |
| (13) | 72.41 | 68.97 | 90.64 | 80.79 | 78.70 |
| (14) | 70.94 | 64.04 | 89.33 | 78.98 | 75.82 |
| (15) | 70.11 | 66.52 | 88.51 | 81.12 | 76.57 |
| (16) | 69.13 | 63.05 | 87.82 | 79.31 | 74.83 |
| (17) | 63.05 | 63.95 | 85.41 | 77.01 | 72.86 |
| (18) | 70.94 | 66.17 | 87.68 | 79.81 | 76.65 |
| (19) | 72.38 | 61.58 | 86.86 | 78.33 | 74.29 |
| (20) | 74.87 | 65.18 | 92.12 | 78.81 | 77.25 |
| CNN + CFC | 0.8211 | **0.7258** | 0.9445 | 0.8376 | 0.8323 |
| DINOv2 | **0.8293** | 0.7145 | 0.9464 | 0.8313 | 0.8304 |
| CNN | 0.8274 | 0.7176 | 0.9367 | 0.8148 | 0.8241 |
| Liquid Dino | **0.8293** | 0.7243 | **0.9503** | **0.8476** | **0.8379** |

Table 3 presents the QC SA8255P NPU inference times (in milliseconds) for the four best performing models. As shown, the Liquid Dino model has the slowest inference time at 8.0 ms, followed by DINOv2 at 7.7 ms, while CNN+CFC and CNN are significantly faster at 3.1 ms and 1.6 ms, respectively.

Despite Liquid Dino being the slowest model in the comparison, it still operates well within the limits required for real-time performance. A processing time of 8.0 ms per frame equates to a potential frame rate of around 125 frames per second, which is more than sufficient for real-time applications in edge devices or automotive systems. Given that real-time systems generally aim for at least 30 frames per second, Liquid Dino's performance comfortably meets the requirements, making it a viable option for deployment in resource-constrained environments such as cars, where both accuracy and response time are critical for monitoring driver behavior.

Table 3: NPU Inference Time for Different Models (in ms)

| Model | QC SA8255P NPU Time (ms) |
|---|---|
| CNN + CFC | 3.1 |
| DINOv2 | 7.7 |
| CNN | 1.6 |
| Liquid Dino | 8.0 |

The set of confusion matrices presented captures the performance of a multi-class classification model across four distinct categories: Emotion, Driver Behavior, Scene-Centric Context and Vehicle-Based Context. These matrices indicate areas of strength in the model's predictive capability, while also highlighting where improvements are necessary, particularly in distinguishing between certain similar classes.

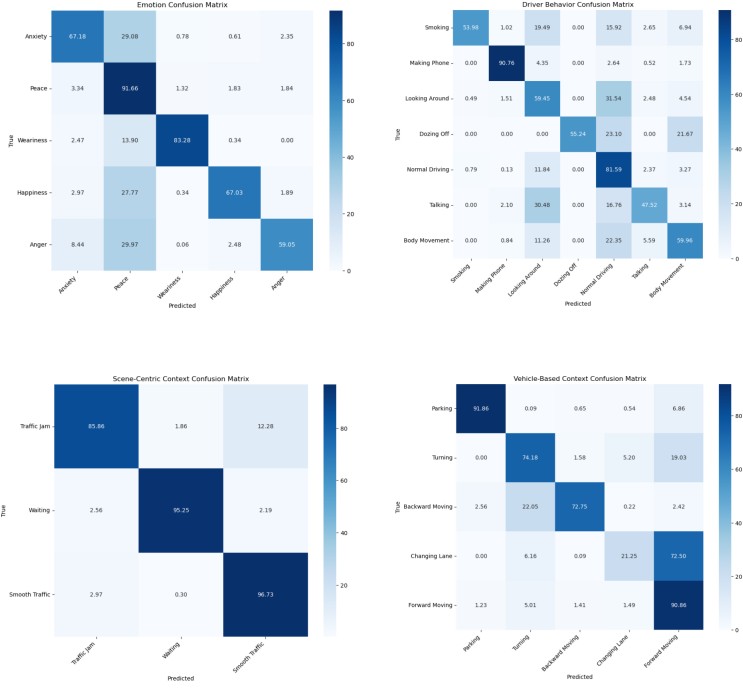

Figure 3: Confusion Matrices for the 4 tasks of Liquid Dino

## 5.5 OVERALL IMPLICATIONS

Our findings highlight the efficacy of advanced neural network architectures in enhancing the accuracy and reliability of multi-task classification in autonomous driving scenarios. By improving the recognition of driver emotions, behaviors and contextual environments, these models contribute

significantly to the development of safer and more efficient autonomous vehicles. Future research can further explore hybrid architectures and real-time implementation strategies to advance the capabilities of autonomous driving systems. Our best model surpasses previous methodologies by at least 5%.

The experimental results clearly demonstrate the superior performance of the Liquid Dino approach. When benchmarked against models using individual methodologies, Liquid Dino achieved an average accuracy of 83.79%, which is higher than any of the other models tested. This significant performance gain is evident across all four classification tasks, with Liquid Dino achieving the highest accuracy in Traffic Context Recognition (95.03%) and Vehicle Condition Recognition (84.76%). This success can be attributed to the model's ability to effectively integrate the strengths of CNNs, DINOv2 and CFC, while mitigating their individual weaknesses. The combination of these methodologies allows Liquid Dino to excel in complex multi-task scenarios, offering enhanced accuracy and robustness.

Lastly, Liquid Dino represents a substantial advancement in machine learning for autonomous driving tasks. By integrating the best aspects of CNNs, DINOv2 and CFC, this innovative methodology sets a new benchmark for performance and efficiency, particularly in multi-task classification settings. Liquid Dino's ability to deliver superior results across multiple tasks, combined with its reduced computational complexity and faster training times, makes it a compelling solution for real-world applications where both efficiency and scalability are critical. These results underscore the potential of Liquid Dino to drive significant progress in the application of continuous-time neural networks in complex, real-world scenarios, particularly in the autonomous driving domain.

# 6 CONCLUSIONS

This study demonstrates significant advancements in the classification of driver emotions, behaviors and contextual environments using state-of-the-art neural network architectures. The novel Liquid Dino model, along with the DINOv2 and CNN configurations, show substantial improvements over traditional models, achieving average accuracies that surpass previous bibliography at least by 5%. The Liquid Dino architecture achieved an overall average accuracy of 83.79%, marking a clear improvement over the alternative models tested. Specifically, it excelled in Traffic Context Recognition, with an accuracy of 95.03% and Vehicle Condition Recognition, achieving 84.76%. These results emphasize the model's ability to generalize well to complex, real-world scenarios and that the combination of liquid networks, which are nonlinear state-space models (SSMs), with transformer architectures like DINOv2 shows promising potential for future work.

A key finding from this research is that the Liquid Dino model, despite its marginally slower inference time compared to other configurations, operates well within the thresholds required for real-time applications, making it a viable option for integration into resource-constrained systems, such as those in autonomous vehicles. The model's ability to maintain high accuracy without compromising operational efficiency highlights its potential for real-world deployment in advanced driver-assistance systems (ADAS). Additionally, the Liquid Dino model processes input frame by frame, eliminating the need to wait for a sequence of frames before making a decision, as is the case with some previous 3D-based methodologies. This capability not only reduces latency but also allows for more immediate and continuous monitoring, which is crucial for applications like driver behavior analysis and safety interventions in real-time systems.

Furthermore, the improvements achieved in classification accuracy are particularly impactful for the development of intelligent driver monitoring systems. By improving the ability to accurately recognize driver states, such as emotions and behaviors, as well as traffic and vehicle conditions, this model contributes directly to the enhancement of road safety. These advancements are pivotal in reducing driver distractions and ensuring timely interventions, ultimately creating a safer driving environment.

In conclusion, this research demonstrates the superiority of the Liquid Dino architecture in addressing the challenges posed by multi-task learning in autonomous driving scenarios. The findings suggest promising avenues for further research, particularly in refining hybrid architectures and optimizing real-time implementation strategies, paving the way for more efficient and reliable autonomous driving systems.

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
