# OpenReview forum: "Liquid Dino: A Multi-Task Neural Network towards Autonomous Driving"
_ICLR.cc/2025/Conference — ICLR 2025 Conference Withdrawn Submission_

### Official Review · Reviewer_jhZy · 2024-11-01

**Soundness:** 2
**Presentation:** 2
**Contribution:** 2
**Rating:** 3
**Confidence:** 5

**Summary:**

In this paper, the authors develop a method named Liquid DiNO, which uses images to classify emotion recognition, driver behavior recognition, scene-centric context recognition, and vehicle-based context recognition. The framework consists of three parts: the first is DiNOv2, the second includes a CNN backbone, and the third is a CFC module. They experiment on a single dataset containing images from three external cameras and one internal camera. The results are presented in terms of accuracy, with the authors claiming that their proposed method performs well.

**Strengths:**

They are trying to solve an important problem using just images.

**Weaknesses:**

The authors in this work attempt to classify specific driver behaviors using a complex framework. However, the paper requires substantial improvement to be considered for acceptance. Below are some of the main weaknesses:

1. The related work section needs to be expanded to include relevant studies. In the third paragraph, fusion techniques are discussed, but this seems irrelevant as no data fusion is performed in this study.

2.  In Figure 1, the driver’s face is partially obscured by wires, and the eyes are not visible. How can meaningful features be learned with such images?
3. Why are all images combined into a single frame? Wouldn’t using weight-sharing in the encoder allow for a better representation of learning from the images?

4.  What role do the external camera images play in the classification task? Does using three external cameras improve the model’s performance, or would a single forward-facing camera suffice?

5. The methodology section does not present a cohesive description of the framework. The parts are divided into unrelated sections, and the motivation for using the CFC module is unclear.

6.  What is the rationale for including a CNN backbone after DiNOv2?

7.  Table 1 is not discussed in the text, and its purpose is unclear.

8. The authors only report accuracy as the evaluation metric. F1 score and AUC should be included to provide a more comprehensive assessment of the framework's performance.
9. There is no ablation study to support their design choices.

**Questions:**

1. The related work section needs to be expanded to include relevant studies. In the third paragraph, fusion techniques are discussed, but this seems irrelevant as no data fusion is performed in this study.

2.  In Figure 1, the driver’s face is partially obscured by wires, and the eyes are not visible. How can meaningful features be learned with such images?
3. Why are all images combined into a single frame? Wouldn’t using weight-sharing in the encoder allow for a better representation of learning from the images?

4.  What role do the external camera images play in the classification task? Does using three external cameras improve the model’s performance, or would a single forward-facing camera suffice?

5. The methodology section does not present a cohesive description of the framework. The parts are divided into unrelated sections, and the motivation for using the CFC module is unclear.

6.  What is the rationale for including a CNN backbone after DiNOv2?

7.  Table 1 is not discussed in the text, and its purpose is unclear.

8. The authors only report accuracy as the evaluation metric. F1 score and AUC should be included to provide a more comprehensive assessment of the framework's performance.
9. There is no ablation study to support their design choices.
10. It would be good for the reader if you provided layer wise details of your framework.

---

### Official Review · Reviewer_XKcX · 2024-11-03

**Soundness:** 2
**Presentation:** 3
**Contribution:** 2
**Rating:** 3
**Confidence:** 4

**Summary:**

In this paper, the authors propose Liquid DINO to achieve more advanced driver assistant multi task learning. The proposed approach shows better performance on the Places365 dataset.

**Strengths:**

1. The paper is easy to understand and follow.

2. The task focused by the authors are very important to the community.

**Weaknesses:**

1. In the introduction section, the authors mentioned that we need to achieve a more advanced architecture to deal with the complex demands of modern driver monitor system. This motivation is not very convincing. The authors should introduce why Liquid Dino can well overcome these complex challenges.



2. The approach is only verified on one dataset. The generalizability of the model is doubtful. Could the authors extend this approach for video based driver distracted behavior recognition dataset, e.g., DriveACT and DAD?

a. Martin, M., Roitberg, A., Haurilet, M., Horne, M., Reiß, S., Voit, M., & Stiefelhagen, R. (2019). Drive&act: A multi-modal dataset for fine-grained driver behavior recognition in autonomous vehicles. In Proceedings of the IEEE/CVF International Conference on Computer Vision (pp. 2801-2810).

b. Kopuklu, O., Zheng, J., Xu, H., & Rigoll, G. (2021). Driver anomaly detection: A dataset and contrastive learning approach. In Proceedings of the IEEE/CVF Winter Conference on Applications of Computer Vision (pp. 91-100).


3. The performance of Liquid Dino is not very promising compared with DINOv2 baseline. Thereby the conttibution of this proposed new method is doubtful.

4. Lack of implementation details. The authors are suggested to add a separate section to introduce the implementation details.

5. The proposed approach is not novel enough. It seems that the performance gain mainly comes from adding more layers for feature learning after DINOv2 framework.

6. Lack of failure case analysis. The authors are suggested to add some qualitative samples with failure cases to illustrate the limitation of the proposed approach.

**Questions:**

Here are the questions based on the provided weaknesses:

1. In the introduction, the authors mention the need for an advanced architecture to address complex requirements of modern driver monitoring systems. Could the authors elaborate on how Liquid Dino specifically overcomes these challenges to strengthen this motivation?

2. To verify the generalizability of the model, would the authors consider extending the approach to other video-based driver distraction behavior recognition datasets, such as Drive&Act (Martin et al., 2019) and DAD (Kopuklu et al., 2021)?

3. Given that Liquid Dino’s performance does not show a marked improvement over the DINOv2 baseline, could the authors clarify the specific contributions of the proposed method that account for any observed gains?

4. Could the authors add a separate section detailing the implementation to provide clearer insights into the architecture, training settings, and parameters used?

5. The proposed approach appears to derive its performance improvement largely from additional feature learning layers following the DINOv2 framework. Could the authors clarify any novel aspects of Liquid Dino beyond adding layers?

6. Could the authors provide qualitative examples of failure cases to illustrate the limitations of Liquid Dino, as this would help clarify areas where the approach may need improvement?

---

### Official Review · Reviewer_SNzt · 2024-11-04

**Soundness:** 2
**Presentation:** 2
**Contribution:** 2
**Rating:** 3
**Confidence:** 4

**Summary:**

This paper presents "Liquid Dino," a multi-task neural network designed to improve the accuracy of advanced driver-assistance systems (ADAS) by classifying various driver states and contextual driving scenarios. The model addresses four classification tasks: Emotion Recognition, Driver Behaviour Recognition, Scene-Centric Context Recognition, and Vehicle-Based Context Recognition by using the visual data captured through multiple cameras both inside and outside the vehicle. AIDE dataset, a multi-view, multi-modal dataset specifically crafted for autonomous driving research is used to evaluate Liquid Dino against various state-of-the-art models.
The model consists of three components: Convolutional Neural Networks (CNNs) for spatial feature extraction, DINOv2 for self-supervised learning, and Closed-form Continuous-Time Neural Networks (CFC) for temporal processing. The architecture is designed to handle diverse data while maintaining high efficiency, with an overall average accuracy of 83.79%, outperforming other models, especially in Traffic Context Recognition (95.03%) and Vehicle Condition Recognition (84.76%).

The presented approach integrates existing methods, thus limiting its novelty. Moreover, it is evaluated using one dataset so justification/conclusion is questionable for a conference like ICLR. However, it promises that the model performs well within the real-time requirements of automotive systems, making it a promising approach for real-world ADAS applications. Additionally, the model's ability to capture driver emotions, behaviours, and contextual driving environments enhances road safety and driver experience, providing valuable contributions to the development of more reliable and responsive autonomous driving systems.

**Strengths:**

The "Liquid Dino" approach is well-thought, particularly through the multi-task learning for autonomous driving. The model integrates Convolutional Neural Networks (CNNs), DINOv2 (self-supervised learning), and Closed-Form Continuous-Time Neural Networks (CFC) to handle spatial, temporal, and unlabeled data. This hybrid architecture incorporates each component's unique strengths, creating a robust model well-suited for the complex, multi-modal requirements of autonomous driving.

Liquid Dino tackles four tasks simultaneously Emotion Recognition, Driver Behavior Recognition, Scene-Centric Context Recognition, and Vehicle-Based Context Recognition. It reflects the diverse demands of real-world driving. Validated on the AIDE dataset, a multi-view, multi-modal dataset that captures rich contextual data under realistic driving conditions, the model demonstrates strong generalization potential. This comprehensive dataset strengthens the relevance and impact of Liquid Dino’s experimental results.

Achieving an overall accuracy of 83.79% and excelling particularly in Traffic Context Recognition (95.03%) and Vehicle Condition Recognition (84.76%), Liquid Dino outperforms existing models despite having an increased inference time of 8 milliseconds per frame. Its frame-by-frame processing, which avoids the need for sequence-based inputs, contributes to immediate, continuous monitoring, ideal for high-stakes ADAS applications.

The model's capacity to integrate driver behaviour, emotion, and environmental context enhances driver safety and experience, supporting the development of more responsive ADAS and autonomous vehicles.

**Weaknesses:**

The main weakness is the novelty. The approach is the integration of the existing approaches and looks like a technical paper.

The second weakness is the evaluation of the proposed approach on a single dataset does not justify its suitability. There are many datasets available for driver behaviour analysis. The list of datasets can be found in the AIDE paper (Yang et al., ICCV 2023). At least an evaluation on another one or two datasets would have made this paper stronger.

In the Introduction (Section 1), the term "Liquid" is introduced, suggesting the use of Liquid Neural Networks (LNNs). However, the methodology primarily combines CNNs, DINOv2, and CFCs, with minimal discussion on the specific role or implementation of LNNs. For a more accurate portrayal of the architecture’s functionality, further elaboration on LNN integration would be beneficial. This could be addressed by referencing the study, "Liquid Neural Networks: A Novel Approach to Dynamic Information Processing" (https://ieeexplore.ieee.org/document/10466162), which explores LNNs' capabilities in handling dynamic data.

In the Experiments (Section 4), the model's performance is evaluated across tasks that have varying temporal dynamics. However, there is limited discussion on how the model adapts to these differences across tasks, which is crucial in applications where time-dependent accuracy is essential. Explaining the confusion matrix scores (Figure 3) for each task in detail would increase the explainability of the model.

The Results (Section 5) provide inference times, which indicate the model’s performance in real-time scenarios, but omit details on the computational resources required during training and deployment. This information is critical for understanding the model's feasibility in resource-constrained environments.

In the Discussion (Section 6), the model's performance is evaluated using the AIDE dataset. However, the paper lacks exploration into how well the model generalizes to other environmental conditions, such as different weather patterns or diverse geographic settings. Testing across various datasets or environments could address potential generalization issues.

The diagram (Figure 2) lacks detail in DINOv2 and CFC modules, with unlabelled arrows between them and CNN, making data transformations unclear. The CNN module omits critical parameters (e.g., kernel size, stride), while multi-view inputs are positioned too far from DINOv2. Missing data dimensions hinder understanding of transformations, and inconsistent module details (e.g., CFC as a single block) disrupt coherence. The absence of loss function indicators and missing representation of any feature fusion or attention mechanisms further limit completeness.

Lastly, in Future Work (Section 8), potential model enhancements are mentioned, but there is a lack of specific details on scalability. Addressing how the architecture can expand to include additional tasks would clarify its future applicability.

**Questions:**

Justification on the novelty of the paper

Experimental evaluation of another one or two datasets listed in the AIDE paper (Yang et al., ICCV 2023).

In the Introduction (Section 1), Could you clarify how the principles of Liquid Neural Networks are integrated into the model architecture? Specifically, how do these principles influence the adaptability and efficiency of the model in processing temporal data?

In Experiments (Section 4), how does the model handle varying temporal dynamics across the four tasks, especially in high-stakes contexts such as emotion and behaviour recognition? Are there specific adaptations for managing differences in the timing and frequency of events in each classification task?

Given the complexity of the multi-task model, are there specific measures or techniques implemented to make the decision-making process interpretable? For safety-critical applications like ADAS, understanding how the model arrives at its predictions is essential for building trust and accountability.

What future enhancements do you envision for Liquid Dino? Specifically, are there plans to scale the architecture for additional tasks within autonomous driving, such as prediction and planning? How would the current model adapt to these expansions?

---

### Note · Authors · 2024-11-23

I have read and agree with the venue's withdrawal policy on behalf of myself and my co-authors.